computational biology

fomite, touchscreen, disease transmission, stochastic model

**Author for correspondence:**
Andrew Di Battista
e-mail: andrew.di.battista@ultraleap.com

# Modelling disease transmission from touchscreen user interfaces

Andrew Di Battista[1], Christos Nicolaides[2,3,5] and Orestis Georgiou[1,4]

[1]Ultraleap Ltd, Bristol, UK
[2]School of Economics and Management, University of Cyprus, Nicosia, Cyprus
[3]Nireas Research Center, and [4]Department of Electrical and Computer Engineering, University of Cyprus, Nicosia, Cyprus
[5]Initiative on the Digital Economy, MIT Sloan School of Management, Cambridge MA, USA

ADB, 0000-0002-9826-0278

The extensive use of touchscreens for all manner of human–computer interactions has made them plausible instruments of touch-mediated disease transmission. To that end, we employ stochastic simulations to model human–fomite interaction with a distinct focus on touchscreen interfaces. The timings and frequency of interactions from within a closed population of infectious and susceptible individuals was modelled using a queuing network. A pseudo-reproductive number $R$ was used to compare outcomes under various parameter conditions. We then apply the simulation to a specific real-world scenario; namely that of airport self-check-in and baggage drop. A counterintuitive result was that $R$ *decreased* with *increased* touch rates required for touchscreen interaction. Additionally, as one of few parameters to be controlled, the rate of cleaning/disinfecting screens plays an essential role in mitigating $R$, though alternative technological strategies could prove more effective. The simulation model developed provides a foundation for future advances in more sophisticated fomite disease-transmission modelling.

## 1. Introduction

The ubiquitousness of shared touchscreen user interfaces (TUIs) has become apparent in recent years; whether it be a fast-food menu or an airport terminal self-check-in machine. However, their reputation for hygiene has come under scrutiny, predominantly from sensationalized media articles [1–3]. The fact that touchscreens carry pathogens is not, however, in question here; what is yet to be established is if they can transmit enough pathogens to a user so as to cause infection (and if so, which disease?), either due to an isolated TUI

**Figure 1.** Queuing network. Infectious and susceptible people from an initial population pool enter a new location at a rate $\lambda_0$. The location model consists of an *arrival* and *departure* first-in-first-out (FIFO) queue; people in the arrival queues wait to interact with one of a number of TUIs, they then move on to the *departure* queue and (potentially) move on to another location at rate $\lambda_{dep}$. Markov chains of conditional probabilities along with a parameter to set the total number of 'jumps' from one location to another is used to govern the flow of people. The rate at which people move through the queues is dictated by the number of TUIs at each location and the rate parameter, $\lambda_{tui}$.

interaction event, or when thought of as a series of interactions by multiple users, effectively forming a fomite transmission network between them [4].

Fomite transmission refers to the transmission of infectious diseases by objects. More specifically, it refers to the transmission of infectious diseases by pathogens left on objects. One common example of this is how the cold virus can be spread by people sneezing and touching door handles [5]. Modelling of fomite-mediated disease transmission has been undertaken in several settings [6,7], where the authors describe an environmental infection transmission system (EITS) using a system of ordinary differential equations (ODEs) incorporating the dominant parameters: pathogen infectivity, survival/ persistence on surfaces and finger-to-surface (surface-to-finger) transfer rates. Other important parameters include the frequency in which people interact/touch a fomite and how often it is disinfected and cleaned [8]. In [9], the authors review the specific role of biometric fingerprint scanners in the transmission of SARS-CoV2. They reiterate the importance of the parameters incorporated in the EITS in addition to recommending enhanced universal hygiene methods; e.g. hand washing, glove wearing, regular surface cleaning and, importantly, the use of non-contact technologies as an overall alternative.

Estimating reasonable values for each disease model parameter relies on often limited experimental and clinical data. For example, the survival rate of pathogens on surfaces has been a source of some contention [10,11]; quantifying viral particles and bacteria is fundamentally not an exact science [12,13]. This is further complicated by the reporting of pathogen quantities using incompatible measures, e.g. plaque forming units (PFU; viruses), colony forming units (CFU; bacterial, fungal), tissue culture infectious dose, with 50% probability of infection ($TCID_{50}$), viral copies from PCR, etc. Moreover, a model needs to consider human behaviour; self-inoculation (transferring of pathogens onto mucosal membranes e.g. mouth, nose, eyes) occurs when individuals touch their faces with contaminated hands (for the purposes of this paper, 'face' touching refers specifically to theses mucosal membranes). It must be assumed that personal hygiene practices such as hand-washing are not strictly nor universally adhered to.

In this paper, rather than target a specific disease model (e.g. influenza, SARS-CoV2, etc.), we investigate disease transmission from a network of TUIs graphically represented in figure 1. More specifically, we focus on the fundamental parameters that govern fomite-mediated transmission due to TUIs and study how these can affect important observables such as the pseudo reproduction number $R$. To that end, we employ stochastic Monte Carlo simulations as they offer more flexibility and ease in incorporating the large number of parameters versus traditional ODE analysis [7,14–17]. Note that we do not consider air-mediated pathogen spread which can often be the primary mode of transmission and is extensively covered in the scientific literature [18]. With specific regard to TUIs, we do, however, examine the impact of using touchless technologies and alternatives such as computer hand tracking using cameras, proximity sensors, RADAR, mid-air haptics [19] and other 'touch-free' interface solutions and compare their effectiveness with the current leading alternative, i.e. more frequent cleaning/disinfection.

# 2. Material and methods

## 2.1. Population model

This disease transmission model differs from the more traditional SEI paradigm in several important ways. Firstly, it is the TUIs themselves that act as the *repositories* for infectious agents; the *spreading* occurs via interaction with human users. This has an impact on what we consider to be 'infectious' compared with the SEI definition in a closed population. Moreover, from a computational perspective, it means that only the bioburdens of the TUIs require monitoring during simulation. TUIs behave essentially as asymptomatic super-spreaders.

The vast majority of TUI users in this model are defined as susceptible ($S$). These are otherwise 'healthy' individuals who, upon interaction with a contaminated TUI surface, pick up pathogens on their fingers/hands. At this point, they become newly exposed ($E$); these individuals *may* then transfer enough pathogens onto the mucosal membrane regions of their face (e.g. eyes, mouth, nose), i.e. self-inoculate if they are to become infected ($I_2$), i.e. $S \rightarrow E \rightarrow I_2$. On the other hand, if the self-inoculation dose is insufficient for infection then $S \rightarrow E \rightarrow S$ [20].

Becoming infected depends largely on the self-inoculation dose and the infectivity of the pathogen under consideration. Keeping track of $I_2$ in our simulations, while varying parameters such as pathogen infectivity or survival rates can gain us insights into what types of pathogens may be most well adapted to fomite transmission.

Becoming *infected* does not subsequently make someone *infectious*. For example, consider an enteric virus such as norovirus. A newly infected individual may take hours/days to become symptomatic and begin shedding large numbers of pathogens. This is unlikely to play a role in our simulations as one can assume that the original TUI location e.g. a shop, restaurant etc. will have undergone a thorough clean overnight (effectively resetting the simulation parameters). Therefore, we do not consider incubation periods or recovery rates. However, we do need to consider the *prevalence* of such individuals in a population in order to answer another crucial question: how do TUIs become contaminated in the first place?

Here, we create another type of individual in our model; the *infectious* donor ($I_1$). They are defined as someone who has relatively high initial levels of pathogens on their hands at any given time, e.g. from coughing/sneezing into one's hands or having recently used the toilet without hand-washing afterwards (see §2.7). Because we consider periods no longer than a day, we assume that they *remain infectious throughout the simulation*. We also assume $I_1$ are themselves immune from *further* infection associated with TUI interaction. It is of interest to see how many $I_2$ are generated for each $I_1$ introduced into the simulation (see §2.9).

## 2.2. Role of exposed ($E$) individuals

In our simulation, we do not consider the re-deposition of pathogens onto touchscreens from exposed $E$ individuals from sequential use (only the bioburdens on TUIs are monitored). This was a design decision based largely on effective use of empirical data pertaining to *touch-transfer efficiency asymmetry* (described further in §2.6.1) [21]. Therefore, in this model susceptible ($S$), exposed ($E$) and newly infected ($I_2$) individuals can only *pick up* pathogens from a TUI surface. Deposition of pathogens onto TUIs is carried out by infectious donor ($I_1$) individuals exclusively.

Another consequence is computational savings from simplifications regarding how self-inoculation (and subsequent probability of infection) is computed (see §2.8.). Essentially, the exposed state is a transitory place-holder: an exposed individual will either self-inoculate ($E \rightarrow I_2$) and potentially become infected or revert back to a susceptible state ($E \rightarrow S$) after TUI use.

## 2.3. Additional scenario assumptions

When considering shared TUIs such as those found in public spaces, we make some further general assumptions which are manifested in our stochastic simulation model:

— Individuals will use the TUI in sequence (i.e. they behave as if in a queue).
— Given typical touchscreen menu design, users are obliged to touch the same regions of the screen, e.g. confirmation buttons, on-screen keypads etc. Therefore, regardless of the application or screen size, users are essentially *sharing the same surface area*.
— We can assume that all touch events are carried out with finger tips (possibly just the index finger of the dominant hand).

— For transmission to occur, we assume individuals are *not* washing their hands before/after using the interface.
— In all simulations, we consider a single time period (i.e. a day) using a 1 min time-step.
— The three main actors in our scenario are therefore the *pathogens*, the network of *touchscreens* and the *network of people*, each having its own controlling parameters.

The remainder of this section describes the implementation of this computer model and presents pseudo code where applicable to provide the reader with the best clarity. A summary table listing all relevant parameters and description can be found in appendix, table 1.

## 2.4. Sampling distributions

Monte Carlo simulations generally make use of a variety of sampling distributions in order to model random events. Throughout this paper we use the *rate parameter* $\lambda$ to describe a *Poisson* process and the symbol $p$ to describe its discrete time counterpart, the *Bernoulli* process. When considering random variables sampled over a particular range, $[a, b]$, we make use of the *truncated normal distribution*, denoted $f(x; \mu, \sigma, a, b)$, where $x$ is a random variable with mode $\mu$ and variance $\sigma^2$. Other random variables are sampled from uniform distributions, $U[a, b]$.

## 2.5. Queuing network model

The movement of people is simulated using a system of first-in-first-out (FIFO) queues (figure 1). We begin with an initial population pool of $N = (S + I_1)$ people (we can assume there is an existing disease prevalence $I_1/N$). These individuals leave the pool at a rate $\lambda_0$, and arrive at one (or any) of $L$ *locations*. Each location has an *arrival* and *departure* queue. Arrivals are people who have yet to interact with the TUI, departures are those who have already interacted and are ready to (potentially) move on to another location (into that location's *arrival* queue). *Locations* can be interpreted as places where there is a cluster of identical TUIs, e.g. a kiosk of ATM machines. An establishment may have several locations within it, each with a different number and type of TUIs serving different customer functions.

The arrival queues are depleted at a rate $m_j \lambda_{\text{tui}}$, ($j = 1, 2, \ldots, L$), where $m_j$ is the number of TUIs at the $j^{th}$ location and $\lambda_{\text{tui}}$ is the rate of TUI use, i.e. $1/\lambda_{\text{tui}}$ is the average time interval between TUI use. In our simulations, $\lambda_{\text{tui}}$ is kept constant across all locations so that a TUI is used on average once every 2 min (in general, it is a parameter associated with each individual TUI design). After a TUI interaction, people may stay at that location for some time before moving on, e.g. eating at a fast food restaurant after ordering a meal. The rate at which people *depart* the location is governed by $\lambda_{\text{dep}}$. Subsequent movement of people between locations is controlled via a Markov chain of conditional probabilities and a *number-of-jumps* parameter that sets how many locations a person can visit before being removed from the active simulation. This framework allows for modelling anything from very basic to increasingly elaborate networks of people movements.

## 2.6. Touchscreen model

### 2.6.1. Transfer efficiency asymmetry

Let us define the deposit rate ($\alpha$) as the proportion of pathogens on an infectious finger transferred onto the surface of a fomite. Similarly, we define pick-up rate ($\beta$) as the proportion of pathogens on a fomite that are transferred to the finger of a susceptible person. With regard to fingers and non-porous surfaces (like glass), transfer efficiency has been shown to be asymmetric. From glass-to-finger, pick-up rates, $\beta$, are on the order of $20 \pm 30$ (s.d.)% [22,23], while deposit rates, $\alpha$, are considerably lower, i.e. 5% [24,25] (figure 2). Some key points worth noting about the experiments conducted to ascertain these values: deposit rates were measured by inoculating a finger with a known concentration of pathogens and measuring the amount left behind on a clean surface. Pick-up rates were examined by touching a contaminated surface with a clean finger and measuring the pathogen level on the finger. Because of the assumption in this model that individuals have either contaminated ($I_1$) or clean hands ($S$) this would suggest that we can interpret the transfer efficiency data as a *one-way* or net result, i.e. either pathogens are deposited by an infectious person or picked up by a susceptible, exclusively.

We model transfer rates using truncated normal distributions; default values are depicted in figure 2*a*. This allows for incorporating the mode, $\mu$, and variance, $\sigma^2$, (i.e. uncertainty) from experimental findings described in the literature while incorporating bounding limits, $(a, b)$, (i.e. 0 to 100%).

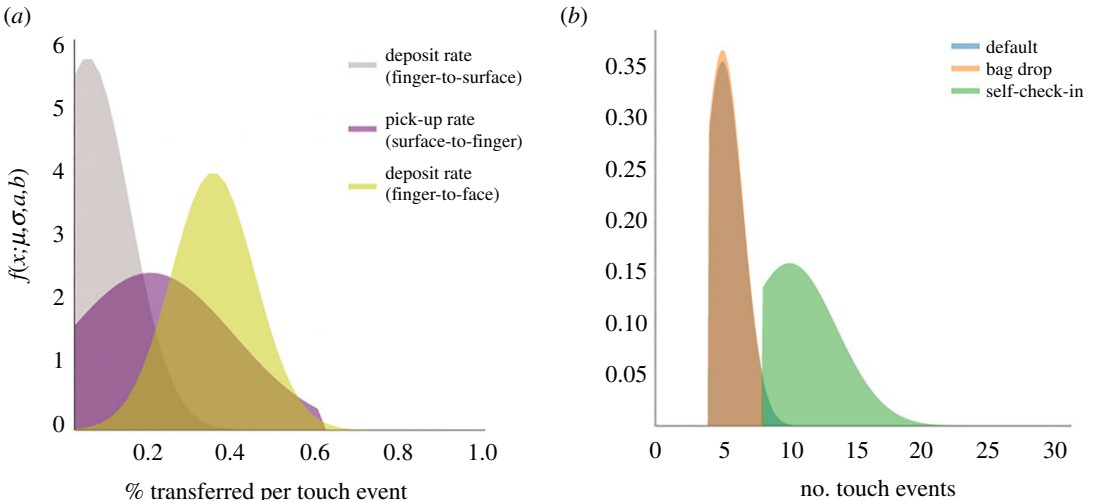

**Figure 2.** Transfer and touch rates. (*a*) Pathogen transfer (pick-up (*β*) and deposit (*α*)) rates are estimated by sampling from truncated normal distributions; default values estimated from the literature are depicted for surface-to-finger, finger-to-surface and finger-to-face. (*b*) Using truncated normal distributions to simulate the average number of touch events during TUI interaction; depicted are a 'default' model used for simulations along with that of a bag-drop and self-check-in machine found at most major airports (note: the 'default' is equivalent to the bag-drop parameters). To inform our choice of distribution parameters, we made use of online video demonstrations of self-check-in procedures; advertised by many major airline companies.

### 2.6.2. Touch rates

As with transfer rates, the number of touches, $n_t$, expected to complete a transaction or menu selection can be modelled using a truncated normal distribution. For example, an ATM pin pad requires a minimum of five touches (four pin numbers + OK), i.e. both mode and minimum can be set to five. However, cancelled transactions, re-attempts, correcting invalid input etc. means that we can expect a small variance in touch numbers. Unless we have compelling reason not to, the maximum number of touches can be safely limited to some reasonably large value, e.g. 30. Figure 2*b* depicts the distributions sampled from to model touchscreen events in this simulation; a default *generic* TUI $f(n_t;$ 5, 1.5, 4, 30), an airport *self-check-in machine* $f(n_t;$ 10, 3.5, 8, 30) and *bag-drop* kiosk $f(n_t;$ 5, 1.5, 4, 8).

### 2.6.3. Pathogen removal

Assuming effective cleaning practices, a thorough wipe with an appropriate cleaning agent will remove approximately 98% of pathogens [26]. We simulate cleaning events with a daily frequency $p_{clean}$. This removal of pathogens is in conjunction with deactivation/die-off rates of bacteria and viruses which can live on surfaces for several hours to months [27,28]. We model this using the pathogen half-life on a surface, $t_{1/2}$. These parameters are likely to play an important role if they are of the same order of magnitude as the intervals between TUI use.

### 2.6.4. Touchscreen dynamical model

Let $D$ be the total accumulated level (or dose) of pathogens on a TUI. Algorithm 1 then describes the flow of pathogens at each simulation time-step. Note, TUI interaction (and all the calculations that accompany it) happens in a single simulation time-step.

In the case of an infectious user, $d_{finger}$ is the number of pathogens *on* their finger. For susceptible users, $d_{pickup}$ is the number of pathogens transferred *to* their finger. For each interaction, we sample a random number of touch events $n_t$ and select a $β$ or $α$ rate from its respective distribution. Note, we only select a transfer rate once per interaction. Here, we assume that an individual will be consistent in the pressure applied with their fingers with each touch (and small variations are essentially incorporated in the random sampling of $β$ and $α$), so transfer rates can be taken as constant throughout the interaction.

The functions *isAvailable()* checks if a TUI is free to use based on rate parameter $\sim Pois(\lambda_{tui})$ (table 1 and figure 1). At this point, an individual at the front of the queue (if not already empty) waiting to use the TUI is selected for interaction.

There are two functions presented in Algorithm 1 that still have yet to be defined *selfInoculate()* and *getPathogenLevelOnFinger()* which will be discussed in the following subsections.

## 2.7. Pathogen shedding

In our model, we would like to estimate the pathogenic load on a *single finger* by considering the following scenarios:

1. After toilet use taking into account hand washing rates and effectiveness; we may consider both respiratory and enteric viruses and bacteria.
2. Coughing and/or sneezing into one's hands assuming a respiratory virus, e.g. influenza.

Viral loads in faeces can be as high as $10^7-10^8$ PFU $\mathrm{g}^{-1}$ [20,29]; this can include common respiratory viruses like influenza in addition to enteric disease causing pathogens. The level of norovirus in faeces has been reported as $10^5-10^9$ particles $\mathrm{g}^{-1}$ based on PCR [30]. Bacterial load found on hands after toilet use ranged from $0.85 \pm 0.93$ (s.d.) $\times 10^5$ CFU for washed and dried hands to $3.64 \pm 4.49$ (s.d.) $\times 10^5$ CFU for unwashed hands [31]. In other experiments, $10^8$ CFU $\mathrm{g}^{-1}$ has been used to approximate natural bacterial contamination levels [32].

It has been estimated that 30% of individuals do not wash their hands sufficiently [33] and there are additional issues in lavatories with regards to using contaminated soap [34] and doorknobs [35]. Therefore, we can feel justified in our assumptions about the prevalence of *infectious* individuals ($I_1$) in a given population.

---

**Algorithm 1** Calculate flow of pathogens in time-step $\Delta t$

...

**for** each time step $\Delta t$ **do**
 **for all** TUI **do**

 $i \leftarrow current(\mathrm{TUI})$ $\triangleright$ index of current TUI

 **if** isAvailable(TUI) **and** deQueue(person) **then**

 $\alpha \leftarrow sample(f(\alpha; 0.05, 2, 0, 60))$ $\triangleright$ see figure 2 / appendix, table 1
 $\beta \leftarrow sample(f(\beta; 0.20, 2, 0, 60))$
 $n_t \leftarrow sample(f(n_t; \mu_i, \sigma_i, a_i, b_i))$

 **if** person = Infectious **then**

 $d_{\mathrm{finger}} \leftarrow getPathogenLevelsOnFinger()$
 $D_i \leftarrow D_i + d_{\mathrm{finger}} \times (1 - (1 - \alpha)^{n_t})$

 **else** {person = (Susceptible **or** Infected)}

 $d_{\mathrm{pickup}} \leftarrow D_i \times (1 - (1 - \beta)^{n_t})$
 $D_i \leftarrow D_i - d_{\mathrm{pickup}}$

 **if** selfInoculate($d_{\mathrm{pickup}}$, $\mathrm{ID}_{50}$) **and** person = Susceptible **then**
 $person \leftarrow Infected$
 **end if**

 **end if**
 **end if**

 **if** isTimeToClean($p_{\mathrm{clean}}$, $\Delta t$) **then**
 $D_i \leftarrow D_i \times (1 - 0.98)$ $\triangleright$ 98% cleaning efficiency
 **end if**

 $\gamma = 2^{(-1/t_{1/2})}$ $\triangleright$ convert half-life to decay rate
 $D_i \leftarrow D_i \times (\gamma)^{\Delta t}$

 **end for**
 **end for**
...

---

The average volume of a cough has been reported at between 0.006 and 0.044 ml [6,8]. Sneeze volumes are estimated as 40 times that of a cough. Coughing and sneezing rates of influenza sufferers

are on the order of 12–22 and 5 h$^{-1}$, respectively. The concentration of viral particles in expulsed droplets, based on nasal swabs, ranges of the order of $10^4 - 10^5$ TCID$_{50}$ ml$^{-1}$.

Regardless of the units quoted (TCID, PFU etc.) the *number* of units are ostensibly of similar orders of magnitude. As will be discussed (see §2.8.1), this number will ultimately be normalized relative to the infectivity of the pathogen under consideration.

From [36], we can estimate that a single fingertip represents approximately 1.4% of the hand's surface and shares that proportion of pathogens. Based on all of the above, we estimate the total dose shed from the finger of an infectious individual (prior to each interaction with a TUI) by algorithm 2.

---
**Algorithm 2** getPathogenLevelsOnFinger() subroutine
---
Function getPathogenLevelsOnFinger()

$np \leftarrow sample(U(10^4, 10^6))$
**return**  $np \times 0.014$          ▷ fingertip equals 1.4% of total hand area

EndFunction

---

## 2.8. Self-inoculation

Face touching rates involving direct contact with mucosal membranes (eyes, nose, mouth etc.) have been found to be approximately 15 touches/h [37,38]. We can simulate the number of face touching events $k$ (we consider a 20 min period following TUI interaction) by drawing from a Poisson distribution, Pois($k$; $\lambda$), where $\lambda = 5$ touches every 20 min. Following a face touch, we model the deposit rate of pathogens from finger-to-lip (more generally skin-to skin) at an average rate of 35% [25,37] (see figure 2 and appendix, table 1).

The rationale behind a 20 min inoculation period is motivated by the following: the survival of pathogens on an exposed individual's fingers will be subject to interactions with countless other fomites and surfaces (e.g. wiping hands on clothing, opening doors, use of mobile phones, etc.) as well as from pathogen survival rates on skin that can range from a few minutes to several hours [39–41]. These effects suggest that initial pathogen exposure, beyond a certain time interval (e.g. 20 min), will be mitigated to the point where infection in no longer viable. Put in another way, we assume only the first five face-touch events (on average) contribute significantly to self-inoculation. We may also consider the survival of pathogens in mucus and the speeds at which the innate immune response takes effect (mucosal membranes being either a hostile or protected environment for an opportunistic pathogen) [42,43].

In short, the assumption is if infection occurs, it will need to happen soon after touchscreen interaction.

### 2.8.1. Dose-response

The infective dose ID$^{50}$ is the estimated number of organisms or pathogen particles required to produce an infection with probability a 50% in normal adult humans. Typically, respiratory viruses require a relatively large dose for infection ($10^3 - 10^4$ TCID$_{50}$) [6,8,44]. For many types of bacteria and enteric viruses this can be as low as 10–100 PFU (or CFU) [45,46]. If we interpret these values as estimates for ID$^{50}$, provided we stick with the same units, pathogen levels can effectively be *normalized*. It is customary to model a dose-response using an exponential cumulative distribution function (CDF) [7]. Self-inoculation is therefore calculated by algorithm 3.

---
**Algorithm 3** selfInoculation() subroutine
---
Function selfInoculation($d_{\text{pickup}}, ID_{50}$)

$\alpha \leftarrow sample(f(\alpha; 0.35, 2, 0, 60))$          ▷ see figure 2 / appendix, table 1

$k \leftarrow sample(\text{Pois}(5))$          ▷ assumes five face touches per 20 min

$d_{inoc} \leftarrow d_{\text{pickup}} \times (1 - (1 - \alpha)^k)$

$P \leftarrow 1 - \exp\left(-\ln(2) \times \frac{d_{inoc}}{ID_{50}}\right)$          ▷ dose response

**return** $accept(P)$          ▷ returns **true** with probability P

EndFunction

---

## 2.9. Outcome measures

In fomite-mediated transmission, a *pseudo reproduction number*, $R$ can be defined as the number of susceptible people that the TUI can infect having been contaminated by an infectious person. Thus, $R$ is defined as the ratio of newly infected individuals to the initial number of infectious (equation (2.1)).

$$R = \frac{\text{number of infected}}{\text{number of infectious}} = \frac{I_2}{I_1}. \tag{2.1}$$

Another metric of interest is the *gap* (in terms of number-of-users) between infectious contamination and subsequent susceptible users becoming infected. It seems intuitive that the next susceptible user will be the most likely to become infected. However, we can measure and store this gap from the simulation results to confirm this assertion. Accordingly, the questions we would like to answer are the following:

— What is the probability of becoming infected after using a TUI?
— On average, how many susceptible individuals could become infected as a direct result of a single infectious user over the course of a day, i.e. $R$?
— Which TUI users are getting infected, i.e. what is the time gap between infectious and infected?
— What is the efficacy of frequent cleaning on reducing the probability of infection?

# 3. Results

## 3.1. Overview

In this section, we present the results from two simulated scenarios; default simulation parameters are listed in appendix, table 1. The touch and transfer rates used are those already discussed and depicted in figure 2. Each scenario is simulated over the period of a single day with 1 min time-step resolution. Care needs to be applied when considering parameters such as population $N$, $\lambda_0$, $\lambda_{\text{tui}}$, etc. to ensure that the entire population actually makes it through the simulation during the allotted time. Results for each parameter setting are averaged over 10 000 realizations of a single day period.

1. **Simulation 1**: A location with a single TUI; this allows us to examine the model's sensitivity to initial conditions and parameters such as survival rates, infectious dose, cleaning rates, etc. We also look at the effects of adding extra TUIs at that location.
2. **Simulation 2**: A real-world example involving two TUI locations; airport terminal *check-in machines* followed by *baggage drop*. We use data for London Heathrow (LHR) Terminal 5 [47] along with the assumption that one in four outgoing passengers makes use of those machines.

## 3.2. Simulation 1: a single TUI location

The following figures show the effects on the reproduction number $R$ for varying different simulation parameters; *disease prevalence* (figure 3a), *pathogen survival*, $t_{1/2}$ (figure 3b), *infectivity*, $\text{ID}^{50}$ (figure 3c), the *number of* TUIs (figure 3d), *touch rates* (figure 3e), *cleaning rate*, $p_{\text{clean}}$ (figure 3f) and the effect of *additional locations* (figure 4). For each simulation, we keep all other parameters constant as given in appendix, table 1.

## 3.3. Simulation 2: airport terminal with two TUI locations

For this simulation, we focus on the effects of *cleaning rate*, $p_{\text{clean}}$ (figure 5a) and compare its effectiveness with substituting a proportion of TUIs with a 'touch-free' alternative (figure 5b).

# 4. Discussion

In many cases, the simulation results are intuitive. For example, it is clear that timing plays an important role as the number of TUIs per location (figure 3d), pathogen survival (figure 3b) and the rate of TUI use, $\lambda_{\text{tui}}$, all interact to affect the infection rate. A key feature of our model was to incorporate the random

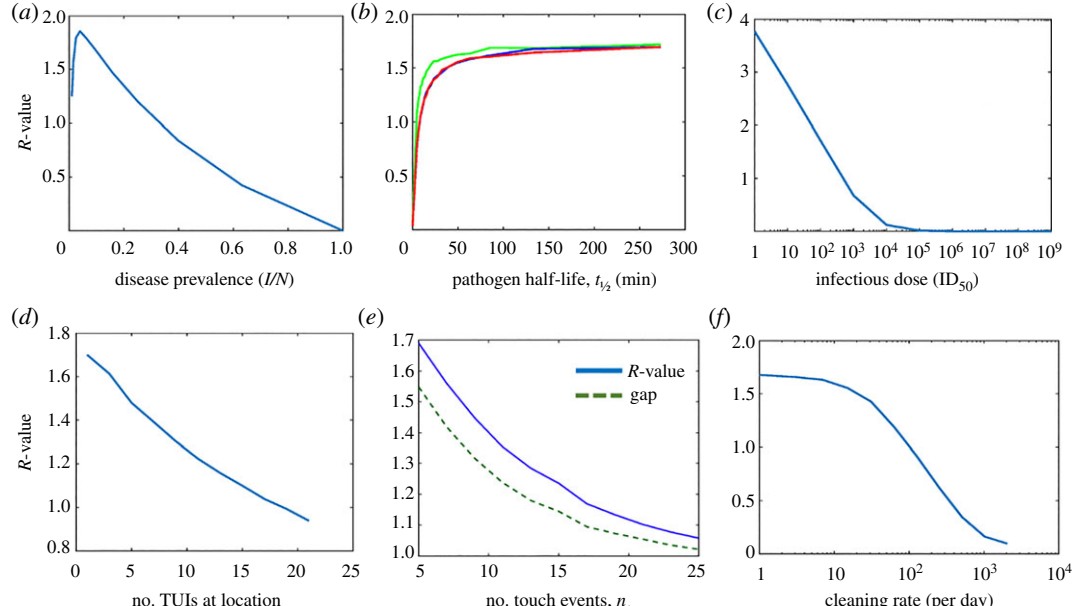

**Figure 3.** Simulation 1 control parameters. (a) Disease prevalence: $R$, reaches a peak at approximately 4%, suggesting an optimal cumulative effect of infectious donor prevalence, beyond which it drops steadily along with the proportion of susceptible individuals. (b) Pathogen survival: a longer half-life results in $R$ asymptotically approaching a maximum value of 1.7. Three separate plots were made using $\lambda_{tui}$ values of 0.5, 0.2 and 0.1, i.e. 2 (green), 5 (blue) and 10 (red) min average interval between TUI use. Longer intervals allow for more time for pathogens to die-off thus slightly lowering the $R$-value. (c) Infectious dose: the effect of varying parameter $ID^{50}$ on $R$ is significant; beyond a certain level of infectivity, fomite-mediated disease transmission becomes non-viable. (d) Number of TUIs: as the number of TUIs available for use at a location increases, the risk of infection drops steadily (approximately linearly). With more TUIs to choose from, the time intervals between their use increases and the effective pick-up and self-inoculation probabilities diminish. (e) Number of touch events: increasing the average number of touches per TUI interaction lowers the infection rates (blue-solid). The average gap (green-dotted) between infectious and infected users indicates that at increasing touch rates the *next* susceptible user of a TUI after its contamination (gap = 1) almost exclusively becomes infected. In other words, a higher touch rate results in a greater pick-up of pathogens, effectively cleaning the surface for subsequent users (shielding them) while simultaneously increasing the probability of infection for the current user. (f) Cleaning rate: cleaning rates, $p_{clean}$, on the order of several hundred times per day are required to achieve an $R$-value less than one. The cleaning intervals are random and are thus not correlated to the rate of TUI users. Therefore, low cleaning rates do not effectively prevent the next susceptible user of the TUI from picking up pathogens.

timing of events through a network queue of people; something that is lost in traditional ODE analysis. Other unsurprising results are the effects of increasing initial disease prevalence (figure 3a) and infectious dose (figure 3c).

Increasing the number of locations (figure 4) essentially gives infectious individuals multiple chances to contaminate TUIs and infect other users. Were it not for the effects of herd-immunity (and the fact that infections were tallied only in the first instance), $R$-values would rise directly proportionally to the number of TUI locations in the simulation. The assumption that an infectious user *always* has some dose of pathogens to deposit at each interaction (algorithm 2), is probably not justified at much higher location numbers as it supposes an inexhaustible shedding of pathogens (or exceptionally unhygienic behaviour). Therefore, figure 4 probably overestimates $R$ as the number of locations increases. Another interpretation would be to say that a 'super-spreader' is necessary for fomite-mediated transmission in the simulated scenarios presented.

Rather less intuitive is the effect of increasing the average number of touches $n_t$ per TUI interaction (figure 3e). Throughout all simulations, the average *gap* recorded between infectious and infected user was between 1 and 2. This implies that the susceptible users who immediately follow an infectious user are most at risk. Because of the asymmetrical way pick-up and deposit rates are modelled, higher touch rates do the same effective job as cleaning the TUI; the next susceptible user is essentially doomed to infection while simultaneously shielding subsequent users.

The TUI is particularly well suited for fomite-mediated disease transmission modelling due to the fact that menu selections can be intuitively incorporated into a simple distribution model (figure 2). Moreover,

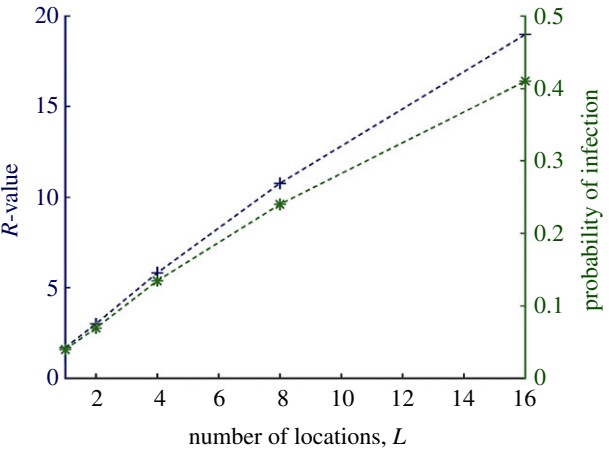

**Figure 4.** Additional locations. Here, we model the effects of additional, identical locations; each simulated person has equal probability of visiting any one location, and will make as many 'jumps' as there are locations. Plotted alongside $R$ (blue +) is the probability of infection (green *), which is calculated as $I_2/S$. Extra locations means more chances for the infectious donor to spread disease. The plots are not precisely linear; the ratio of $R$ to the number of locations *decreases* with *increasing* location numbers. In all simulations, people can only be *infected* once (despite picking up more infectious doses). Therefore, this reduction in infection rate efficiency is due to herd-immunity.

the non-porous surface type means that touch-transfer coefficients relate closely to appreciable empirical data from the literature (which often involve finger to glass/ceramic tile contact).

We did not model the *re-deposition* of pathogens from exposed or newly infected individuals onto TUI fomites due to the complexity and consequent unreliability in estimating pathogen levels on an individual's hand over time. Without re-deposition, pathogens that are picked up are essentially removed from the system. Our model, therefore, underestimates the overall bioburden of the network of TUIs (if we consider the effects of this one phenomenon only). A focus of future work will be to adapt the algorithm to monitor finger bioburden over time and reconfigure the transfer efficiency model to account for a gradient between concentrations of microbes on finger and surface.

An additional future modelling improvement (that would follow from monitoring pathogen levels on the finger/hands) would be an updated self-inoculation model to incorporate a dynamic dose-response, i.e. accounts for dose and survival of pathogens on mucosal membranes. In addition to face-touch events, the touching of clothing, skin and other personal objects would contribute further to increased pathogen removal from the system. This would probably decrease infection rates further.

In order to use our simulation tool to model a specific disease, one would have to collate accurate shedding rates and infectious dose information. This requires care when dealing with TCID$_{50}$, PFU, CFU, etc. Thus, the default parameters used in our simulations were not tuned to a specific pathogen type (but may be attributed to certain strains of *E. coli* or adenovirus). Enteric disease causing pathogens [39,41] ostensibly have the right combination of relatively long half-life on surfaces and low infectious dose to be the major players in fomite-mediated transmission. However, drawing any conclusions about a specific pathogen using this model should be done with caution.

The goal of this paper is to investigate *how* different model parameters affect fomite-mediated disease transmission, including the human factor captured here through parameters that control the movement of people through the TUI location network. With accurate prior information about a specific pathogen and TUI design, one could consult figure 3 and gain useful insight into the possible consequences.

To date, TUI use has not been associated with any major disease outbreak. This could be partly due to the nature of the pathogens one is likely to pick up from such surfaces; many of which do not cause sudden and acute illness, e.g. if an individual develops a mild cough several days after TUI use, they are not likely to report the illness nor associate it with the TUI interaction. The lack of good epidemiological evidence is, therefore, understandable. However, this does not mean that more serious cases do not occur and that it is not in the interest of public health and safety to try and mitigate transmission whenever possible.

In this paper, we made use of a pseudo $R$-value. It should be clear that, while an $R$-value less than 1 is desirable in a pandemic, in this scenario a user-interface designer should be aiming much lower ($R \ll 1$).

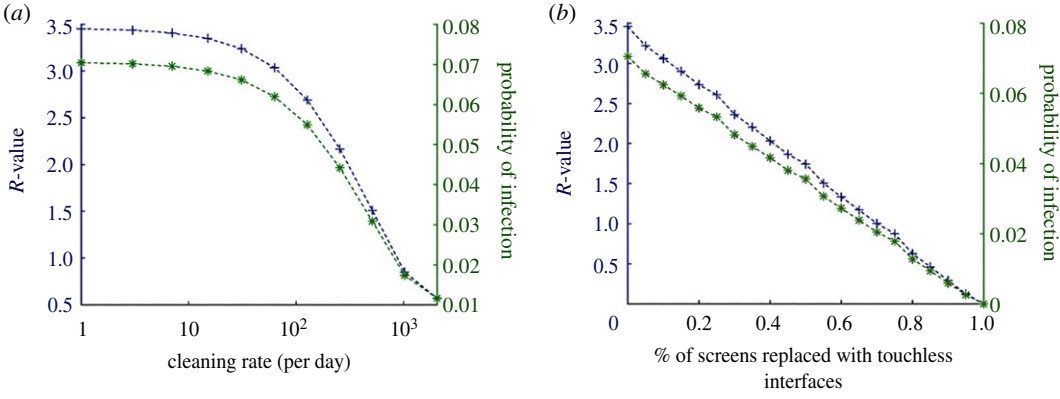

**Figure 5.** Cleaning rate versus 'touch-free' interventions. (*a*) The effects of cleaning are similar to that in Simulation 1; *R*-values are higher overall due to having more than one location. (*b*) Replacing a proportion of TUIs with a 'touch-free' alternative results in a direct linear drop in *R*. For comparison, the same *R*-value is achieved by replacement of 50% of TUIs as a cleaning rate of several hundred cleans per day per TUI, of which there are 60 in this simulation.

One effective way to mitigate infection spread via fomites is with compulsory hand washing. This has also been shown to be effective against pandemics in airport networks [17,48]. Indeed, the relevance of our paper's results depends on whether or not a population will maintain these stringent hygienic practices.

An alternative approach that places the responsibility and control with the TUI owners/operators is enhanced sterilization regimens. From figures 3*f* and 5*a*, it is apparent that cleaning rates of the order of hundreds of times per day per TUI are required to have a significant effect on *R*. From a cost perspective, this can be prohibitive. In addition to the added cost of cleaning agents, protective equipment (e.g. gloves) and increased staff exposure, the excessive use of industrial and household cleaning agents carries with it a health risk, particularly to those with breathing ailments [49,50].

Ultraviolet light is an alternative to chemical agents for disinfecting surfaces, but does not solve the issue of increased cleaning rates nor is it completely absolved from health implications [51–54].

A promising and attractive solution is the emerging technology of self-cleaning antimicrobial surface coatings, many of which are commercially available, e.g. for tablets/smartphones [55–57]. With respect to the model presented in this paper, these coatings would have significant impact on the pathogen half-life parameter, $t_{1/2}$, (figure 3*b*). However, not all pathogens are significantly affected [58]. It is also unclear if their antimicrobial properties diminish with regular use or require maintenance and/or replacement.

Finally, another alternative to mitigate disease transmission from TUIs are touch-free interfaces that completely remove the need to touch and therefore deposit or pick up pathogens from surfaces (figure 5*b*). Touch-free interfaces have long been studied in academic human–computer interaction literature and have been realized in multiple consumer and enterprise products [59,60].

Data accessibility. Data and relevant code for this research work are stored in Gitlab: https://github.com/andydiba/fomite_sim and have been archived within the Zenodo repository: https://doi.org/10.5281/zenodo.5084601.

Competing interests. We declare we have no competing interests.

Funding. O.G. has received funding from the European Union's Horizon 2020 research and innovation programme under the Marie Skłodowska-Curie project NEWSENs, grant agreement no. 787180. C.N. has received funding from the European Union's Horizon 2020 research and innovation programme under the Marie Skłodowska-Curie project NISIHealth, grant agreement no. 786247.

Acknowledgements. This research project was conducted in collaboration with Ultraleap Ltd, and the authors gratefully acknowledge their advice and support.

# Appendix A

(see table 1)

**Table 1.** Simulation parameters. Summary of simulation parameters. The distribution/range column shows the range or default values simulated over to produce figures 3–5. Specific default parameters for Simulations 1 and 2 are listed under Sim. 1 and Sim. 2 columns. Simulations are carried out using 10 000 realizations over a 1 day period (with 1 min time-steps resolution). Simulation 1 models one location with a single TUI. Simulation 2 models an airport terminal with 36 self-check-in machines and 24 bag-drop machines. The figure of $N = 12\,000$ is derived from passenger arrival data from LHR T5 (2018) and assuming one in four passengers actually makes use of the machines [47].

| symbol | description | distribution/range | Sim. 1 | Sim. 2 |
|---|---|---|---|---|
| **pathogen Model** | | | | |
| $n_p$ | number of pathogens shed by an infectious individual | $U(10^4, 10^6)$ | | |
| $ID^{50}$ | infectious dose (50% probability) | $10^0$–$10^9$ | | $10^2$ |
| $t_{1/2}$ | pathogen half-life on surface | 5 min–24 h | | 3 (h) |
| **self-inoculation model** | | | | |
| $k$ | face-touch events per 20 min | $Pois(k; 5)$ | | |
| $\alpha_{face}$ | deposit rate (finger to face) | $f(\alpha_{face}; 0.35, 0.1, 0, 1)$ | | |
| **TUI model** | | | | |
| $\alpha_{tui}$ | deposit rate (tui) | $f(\alpha_{tui}; 0.05, 0.1, 0, 0.6)$ | | |
| $\beta$ | pick-up rate (tui) | $f(\beta; 0.20, 0.2, 0, 0.6)$ | | |
| $n_t$ | number of touches (default) | $f(n_t; 5, 1.5, 4, 30)$ | | |
| | number of touches (self-check-in) | $f(n_t; 10, 3.5, 8, 30)$ | | |
| | number of touches (bag drop) | $f(n_t; 5, 1.5, 4, 8)$ | | |
| $\lambda_{tui}$ | TUI use rate (min$^{-1}$) | 0.5 | | 0.5 |
| $p_{clean}$ | TUI cleaning rate (daily) | 0–1440 | | 0 |
| **queuing network model** | | | | |
| L | number of locations | 1–2 | 1 | 2 |
| $m_i$ | TUIs per location | 1–36 | 1 | (36, 24) |
| N | population | 100–12 000 | 100 | 12 000 |
| $I_1/N$ | disease prevalence | 0–100% | | 2% |
| $\lambda_0$ | initial arrival rate (min$^{-1}$) | 0.5–20 | 0.5 | 20 |
| $\lambda_{dep}$ | departure rate (min$^{-1}$) | 0.05–1000 | 0.05 | 1000 |

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
