## [Peer Review File · Royal Society Open Science]

Review History

RSOS-210625.R0 (Original submission)

Review form: Reviewer 1 (Jose Jimenez)

Is the manuscript scientifically sound in its present form?

Yes

Are the interpretations and conclusions justified by the results?

Yes

Is the language acceptable?

Yes

Do you have any ethical concerns with this paper?

No

Have you any concerns about statistical analyses in this paper?

No

Recommendation?

Accept as is

Comments to the Author(s)

General information:

By definition, a simulation of a dynamic disease model is complex to say the least. The decision to review the process of touchscreens and their potential to infect a population is very relevant, especially during a time of pandemic. I believe that your method to tackle the problem at hand was an elegant one. I also believe that you have the potential to extend your model into other types of fomites, and additional populations.

Additional comments:

-Your paper utilizes a combination of queuing and SEIS model. I believe that this greatly allows to show the movement of people in a more realistic manner compared to a simple ODE or a systems dynamics model. One way to improve this in the future, would be to map the motion of people through an airport terminal or restaurant.

-By using C you greatly increased the flexibility of your model. Any change to the infection rates can be included easily or could even be varied in a parameter input table that feeds to your model. This could greatly increase the possibility of multi-scenario simulation.

-I was very impressed by the amount of initial information that the team researched for verification and validation of the model.

-In the future, a more heterogeneous population that shows different demographics would be interesting for more detailed studies. For example, the difference between children, young adults, adults, and senior citizens.

Excellent work on this study!

Review form: Reviewer 2

Is the manuscript scientifically sound in its present form?

Yes

Are the interpretations and conclusions justified by the results?

No

Is the language acceptable?

Yes

Do you have any ethical concerns with this paper?

No

Have you any concerns about statistical analyses in this paper?

Yes

Recommendation?

Major revision is needed (please make suggestions in comments)

Comments to the Author(s)

The article presents a study for modeling the transmission of infectious disease through touchscreen user interfaces (TUIs), within a closed population that interacts with a network of TUIs.

My main concern regarding this article is the presentation of the model. Except for a graphical description of the TUIs network, the article is missing a more formal and detailed description of the population dynamics model: how individuals in the study population change their health states (from susceptible to infectious and to infected)? This is in my opinion an important point, as directly linked to the main outcome measure R. The algorithms pseudocode provided in the paper describes the evolution of the TUIs state in terms of cumulated pathogens dose, but what about the evolution of the population state? How the function 'deQueue()' and isAvailable() are defined?

As a reader, I was not able to understand what the authors mean by susceptible, infectious, infected, and exposed individuals. These are well-established epidemiological concepts but the way they are used in this study is misleading.

The authors defined an infectious individual as 'someone who has relatively high initial levels of pathogens on their hands at any given time', then is the individual infected too?

An exposed individual is someone who's contaminated hands after the interaction with a TUI, then is the individual infectious too?

The authors should clearly state their definitions somewhere in the model description.

One of the model parameters is the disease prevalence, how is it actually used in the model? I suppose to initialize the population for the simulations.

The prevalence of a disease is the proportion of a population infected at a given point in time. And the population under study is a population of infectious and susceptible individuals. But according to the authors' definition of infectious individuals (i.e., individuals with contaminated hands), infected individuals are not necessarily infectious. How do you deal with this?

The model builds on a number of assumptions. Some of them sound questionable and would require further investigation.

Authors state that infected individuals 'remain infectious throughout the simulation' and that a simulation lasts one day. I find this assumption implausible.

Similarly, the self-inoculation timing of 20 min seems excessive as the effects of interactions with other surfaces are not negligible for such a long time window.

Also, assuming susceptible, exposed, and newly infected individuals cannot deposit pathogens onto TUIs is a strong assumption, specific to the pathogens' nature and thus cannot be generalized.

Concerning the simulation parameters values, the authors are reporting 2 tables (Table 1 and Table A1), I would merge the 2.

Results analysis is weak and would merit a further discussion.

To conclude, I think that the authors have proposed a very general framework for modeling the transmission of infectious diseases through TUIs, presenting several unclear and uncertain points that would need further consideration. At this stage, no practical conclusions and/or recommendations can be drawn from it.

Thus conclusions such as 'Results revealed that the required rate of cleaning/disinfecting of screens to effectively mitigate R can be inordinately high. This suggests that revised strategies or alternative methods should be considered.' sound not appropriate.

Decision letter (RSOS-210625.R0)

Dear Dr Di Battista

The Editors assigned to your paper RSOS-210625 "Modelling disease transmission from touchscreen user interfaces" have now received comments from reviewers and would like you to revise the paper in accordance with the reviewer comments and any comments from the Editors. Please note this decision does not guarantee eventual acceptance.

Please submit your revised manuscript and required files (see below) no later than 21 days from today's (ie 21-Jun-2021) date. Note: the ScholarOne system will 'lock' if submission of the revision is attempted 21 or more days after the deadline. If you do not think you will be able to meet this deadline please contact the editorial office immediately.

on behalf of Professor Enrico Bertuzzo (Associate Editor) and Mark Chaplain (Subject Editor)
openscience@royalsociety.org

Associate Editor Comments to Author (Professor Enrico Bertuzzo):

Associate Editor: 1

Comments to the Author:

The manuscript has been reviewed by two referees. One review is positive. The other one also finds the material interesting and sees the potentialities, but also raises some concerns which I partially share. Although I found the description of the model less unclear than this referee

found, it could certainly be improved for readability and clarity. For instance stating very clearly at the beginning that infectious individuals (I1) does not change during the simulations, and that infected individuals (I2) do not become infectious during the simulation horizon (it should also be stressed that the latter is disease dependent). This is different from the usual epidemiological model that runs over longer simulation times and sees a flux $I \rightarrow S \rightarrow E \rightarrow I$, and it is worth emphasizing.

I found neglecting re-deposition of pathogens a strong and difficult to justify assumption. Given the framework that authors developed, it should be quite simple to accommodate this additional process.

I share the reviewer assessment that the discussion and conclusions sections could be further developed. The relevance of the results is overstated given the set of hypotheses and should be suitably toned down.

Reviewer comments to Author:

Reviewer: 1

Comments to the Author(s)

General information:

By definition, a simulation of a dynamic disease model is complex to say the least. The decision to review the process of touchscreens and their potential to infect a population is very relevant, especially during a time of pandemic. I believe that your method to tackle the problem at hand was an elegant one. I also believe that you have the potential to extend your model into other types of fomites, and additional populations.

Additional comments:

-Your paper utilizes a combination of queuing and SEIS model. I believe that this greatly allows to show the movement of people in a more realistic manner compared to a simple ODE or a systems dynamics model. One way to improve this in the future, would be to map the motion of people through an airport terminal or restaurant.

-By using C you greatly increased the flexibility of your model. Any change to the infection rates can be included easily or could even be varied in a parameter input table that feeds to your model. This could greatly increase the possibility of multi-scenario simulation.

-I was very impressed by the amount of initial information that the team researched for verification and validation of the model.

-In the future, a more heterogeneous population that shows different demographics would be interesting for more detailed studies. For example, the difference between children, young adults, adults, and senior citizens.

Excellent work on this study!

Reviewer: 2

Comments to the Author(s)

The article presents a study for modeling the transmission of infectious disease through touchscreen user interfaces (TUIs), within a closed population that interacts with a network of TUIs.

My main concern regarding this article is the presentation of the model. Except for a graphical description of the TUIs network, the article is missing a more formal and detailed description of the population dynamics model: how individuals in the study population change their health states (from susceptible to infectious and to infected)? This is in my opinion an important point, as directly linked to the main outcome measure R. The algorithms pseudocode provided in the paper describes the evolution of the TUIs state in terms of cumulated pathogens dose, but what

about the evolution of the population state? How the function 'deQueue()' and isAvailable() are defined?

As a reader, I was not able to understand what the authors mean by susceptible, infectious, infected, and exposed individuals. These are well-established epidemiological concepts but the way they are used in this study is misleading.

The authors defined an infectious individual as 'someone who has relatively high initial levels of pathogens on their hands at any given time', then is the individual infected too?

An exposed individual is someone who's contaminated hands after the interaction with a TUI, then is the individual infectious too?

The authors should clearly state their definitions somewhere in the model description.

One of the model parameters is the disease prevalence, how is it actually used in the model? I suppose to initialize the population for the simulations.

The prevalence of a disease is the proportion of a population infected at a given point in time.

And the population under study is a population of infectious end susceptible individuals. But according to the authors' definition of infectious individuals (i.e., individuals with contaminated hands), infected individuals are not necessarily infectious. How do you deal with this?

The model builds on a number of assumptions. Some of them sound questionable and would require further investigation.

Authors state that infected individuals 'remain infectious throughout the simulation' and that a simulation lasts one day. I find this assumption implausible.

Similarly, the self-inoculation timing of 20 min seems excessive as the effects of interactions with other surfaces are not negligible for such a long time window.

Also, assuming susceptible, exposed, and newly infected individuals cannot deposit pathogens onto TUIs is a strong assumption, specific to the pathogens' nature and thus cannot be generalized.

Concerning the simulation parameters values, the authors are reporting 2 tables (Table 1 and Table A1), I would merge the 2.

Results analysis is weak and would merit a further discussion.

To conclude, I think that the authors have proposed a very general framework for modeling the transmission of infectious diseases through TUIs, presenting several unclear and uncertain points that would need further consideration. At this stage, no practical conclusions and/or recommendations can be drawn from it.

Thus conclusions such as 'Results revealed that the required rate of cleaning/disinfecting of screens to effectively mitigate R can be inordinately high. This suggests that revised strategies or alternative methods should be considered.' sound not appropriate.

===PREPARING YOUR MANUSCRIPT===

===PREPARING YOUR REVISION IN SCHOLARONE===

- Ensure that your data access statement meets the requirements at <https://royalsociety.org/journals/authors/author-guidelines/#data>. You should ensure that you cite the dataset in your reference list. If you have deposited data etc in the Dryad repository, please include both the 'For publication' link and 'For review' link at this stage.
- If you are requesting an article processing charge waiver, you must select the relevant waiver option (if requesting a discretionary waiver, the form should have been uploaded at Step 3 'File upload' above).
- If you have uploaded ESM files, please ensure you follow the guidance at <https://royalsociety.org/journals/authors/author-guidelines/#supplementary-material> to include a suitable title and informative caption. An example of appropriate titling and captioning may be found at https://figshare.com/articles/Table_S2_from_Is_there_a_trade-off_between_peak_performance_and_performance_breadth_across_temperatures_for_aerobic_scope_in_teleost_fishes_/3843624.

Author's Response to Decision Letter for (RSOS-210625.R0)

See Appendix A.

Decision letter (RSOS-210625.R1)

Dear Dr Di Battista

On behalf of the Editors, we are pleased to inform you that your Manuscript RSOS-210625.R1 "Modelling disease transmission from touchscreen user interfaces" has been accepted for publication in Royal Society Open Science subject to minor revision in accordance with the Editors' comments below my signature.

We invite you to respond to the comments and revise your manuscript. Below the Editors' comments we provide additional requirements. Though further review of the manuscript is unlikely to be necessary after you prepare and resubmit the manuscript, final acceptance is dependent on these requirements being met. We provide guidance below to help you prepare your revision.

Please submit your revised manuscript and required files (see below) no later than 7 days from today's (ie 08-Jul-2021) date. Note: the ScholarOne system will 'lock' if submission of the revision

is attempted 7 or more days after the deadline. If you do not think you will be able to meet this deadline please contact the editorial office immediately.

on behalf of Professor Enrico Bertuzzo (Associate Editor) and Mark Chaplain (Subject Editor)
openscience@royalsociety.org

Associate Editor Comments to Author (Professor Enrico Bertuzzo):

The authors revised the manuscript convincingly and it can be accepted (almost) as is.

The last 5 rows of the manuscript are out of the scope of the paper and I am asking the authors to delete them when preparing the final files. Moreover, please note that all references are gone in the version I am reading.

Pag 2, line 51 "On the other hand"

===ZENODO DEPOSITION===

At this stage, we ask that you please archive your GitHub code within the Zenodo repository: <https://guides.github.com/activities/citable-code/>. By doing this, a formal, citable DOI will be associated with your data record, and an open license (CC-BY preferred) can be applied to your data. We would then ask that you please update your data availability statement to read as:

"Data and relevant code for this research work are stored in GitHub: [GitHub URL here] and have been archived within the Zenodo repository: <https://doi.org/zenodo.....> [ref number].

===PREPARING YOUR MANUSCRIPT===

Your revised paper should include the changes requested by the referees and Editors of your manuscript. You should provide two versions of this manuscript and both versions must be provided in an editable format:
one version identifying all the changes that have been made (for instance, in coloured highlight, in bold text, or tracked changes);
a 'clean' version of the new manuscript that incorporates the changes made, but does not highlight them. This version will be used for typesetting.

===PREPARING YOUR REVISION IN SCHOLARONE===

- If you are providing image files for potential cover images, please upload these at this step, and inform the editorial office you have done so. You must hold the copyright to any image provided.
- A copy of your point-by-point response to referees and Editors. This will expedite the preparation of your proof.

- Ensure that your data access statement meets the requirements at <https://royalsociety.org/journals/authors/author-guidelines/#data>. You should ensure that you cite the dataset in your reference list. If you have deposited data etc in the Dryad repository, please only include the 'For publication' link at this stage. You should remove the 'For review' link.
- If you are requesting an article processing charge waiver, you must select the relevant waiver option (if requesting a discretionary waiver, the form should have been uploaded at Step 3 'File upload' above).
- If you have uploaded ESM files, please ensure you follow the guidance at <https://royalsociety.org/journals/authors/author-guidelines/#supplementary-material> to include a suitable title and informative caption. An example of appropriate titling and captioning may be found at https://figshare.com/articles/Table_S2_from_Is_there_a_trade-off_between_peak_performance_and_performance_breadth_across_temperatures_for_aerobic_scope_in_teleost_fishes_/3843624.

Author's Response to Decision Letter for (RSOS-210625.R1)

See Appendix B.

Decision letter (RSOS-210625.R2)

Dear Dr Di Battista,

I am pleased to inform you that your manuscript entitled "Modelling disease transmission from touchscreen user interfaces" is now accepted for publication in Royal Society Open Science.

You can expect to receive a proof of your article in the near future. Please contact the editorial office (openscience@royalsociety.org) and the production office (openscience_proofs@royalsociety.org) to let us know if you are likely to be away from e-mail contact – if you are going to be away, please nominate a co-author (if available) to manage the proofing process, and ensure they are copied into your email to the journal. Due to rapid publication and an extremely tight schedule, if comments are not received, your paper may experience a delay in publication.

on behalf of Professor Enrico Bertuzzo (Associate Editor) and Mark Chaplain (Subject Editor)
openscience@royalsociety.org

Appendix A

Modelling disease transmission from touchscreen user interfaces

Associate Editor 1:

The manuscript has been reviewed by two referees. One review is positive. The other one also finds the material interesting and sees the potentialities, but also raises some concerns which I partially share. Although I found the description of the model less unclear than this referee found, it could certainly be improved for readability and clarity. For instance, stating very clearly at the beginning that infectious individuals (I1) does not change during the simulations, and that infected individuals (I2) do not become infectious during the simulation horizon (it should also be stressed that the latter is disease dependent). This is different from the usual epidemiological model that runs over longer simulation times and sees a flux $I \rightarrow S \rightarrow E \rightarrow I$, and it is worth emphasizing.

We thank the Associate Editor for the positive comments and constructive feedback.

We have followed the advice provided and included statements clarifying our model and note the differences from the usual epidemiological models, thus improving readability of our revised manuscript.

I found neglecting re-deposition of pathogens a strong and difficult to justify assumption. Given the framework that authors developed, it should be quite simple to accommodate this additional process.

We have found that accommodation re-deposition does require some significant change to the way in which pick-up and deposit rates are interpreted (as these are derived from experiments where either finger or surface is initially clean). From a computational perspective, this also requires some considerable changes to data structures to accommodate monitoring bioburdens of each individual. While we agree that it is certainly 'doable', such modifications are perhaps better presented as an update to the current framework, which firstly provides a foundation a network of TUI interactions.

In terms of the influence on infection risk or 'R', re-deposition essentially keeps pathogens 'alive' in the simulation that are otherwise being removed in the current framework. Assuming more pathogens equals higher risk then our current framework may underestimate infection risk (though in reality the implications will be more complex).

I share the reviewer assessment that the discussion and conclusions sections could be further developed. The relevance of the results is overstated given the set of hypotheses and should be suitably toned down.

We have carefully followed the reviewers' comments and included further discussion points in the conclusion section to address those. We have also revised the wording relating to our results and their relevance as advised.

We hope that these modifications and our responses to the reviewers' comments (see point-by-point response below) are sufficient and acceptable for the publication of our paper at your journal.

Reviewer 1:

General information:

By definition, a simulation of a dynamic disease model is complex to say the least. The decision to review the process of touchscreens and their potential to infect a population is very relevant, especially during a time of pandemic. I believe that your method to tackle the problem at hand was an elegant one. I also believe that you have the potential to extend your model into other types of fomites, and additional populations.

We thank the Reviewer positive comments and constructive feedback and are keen to extend our results in future work.

Additional comments:

-Your paper utilizes a combination of queuing and SEIS model. I believe that this greatly allows to show the movement of people in a more realistic manner compared to a simple ODE or a systems dynamics model. One way to improve this in the future, would be to map the motion of people through an airport terminal or restaurant.

-By using C you greatly increased the flexibility of your model. Any change to the infection rates can be included easily or could even be varied in a parameter input table that feeds to your model. This could greatly increase the possibility of multi-scenario simulation.

-I was very impressed by the amount of initial information that the team researched for verification and validation of the model.

-In the future, a more heterogeneous population that shows different demographics would be interesting for more detailed studies. For example, the difference between children, young adults, adults, and senior citizens.

Excellent work on this study!

We agree with the reviewer and also thank him/her for the great suggestions. In future work we plan to look at specific case studies and create this kind of flow-maps while also looking at different personas and demographics.

Reviewer 2:

The article presents a study for modelling the transmission of infectious disease through touchscreen user interfaces (TUIs), within a closed population that interacts with a network of TUIs.

My main concern regarding this article is the presentation of the model. Except for a graphical description of the TUIs network, the article is missing a more formal and detailed description of the population dynamics model: how individuals in the study population change their health states (from susceptible to infectious and to infected)? This is in my opinion an important point, as directly linked to the main outcome measure R.

We thank the Reviewer positive comments and constructive feedback.

We have followed the advice provided and revised our manuscript to clarify our model. Changes are marked in Blue (new text) and Red (removed old text). Specifically, we have defined the population dynamics and clarified our assumptions in bullet points in the relevant subsections (see pages 2-4). Briefly, we make a clear distinction between the definitions of 'infectious' and 'infected' and further clarify the transitions between susceptible, exposed and infected states.

The algorithms pseudocode provided in the paper describes the evolution of the TUIs state in terms of cumulated pathogens dose, but what about the evolution of the population state? How the function 'deQueue()' and isAvailable() are defined?

The queues to arrive/leave locations and use a TUI are controlled by Poisson process parameters (denoted by λ) and listed in the table of parameter (Table A1) in the appendices and described in Fig1. For example, $1/L\{tui\}$ is the average time for a TUI interaction, which is set to a default of 2 minutes. The pseudo code essentially attempts to convey that (at a particular time instant) if a TUI is available and a person is waiting in the queue to use it, then they do. More technically, `deQueue(person)` loads a person (if the queue is not already empty) from the queue into a temporary structure ('person') that is used for subsequent computations and updates.

We have added an additional paragraph after 'Algorithm 1' to summarize these functions.

As a reader, I was not able to understand what the authors mean by susceptible, infectious, infected, and exposed individuals. These are well-established epidemiological concepts but the way they are used in this study is misleading.

We thank the reviewer for highlighting this problem. We have addressed this problem in our revised manuscript by clarifying their definitions in our population and SEIS model in the form of bullet points and examples. We note that we have not deviated from the well-established epidemiological concepts but do expand the infected group into newly infected I2 and originally infectious I1. We note that we are dealing with a conservative system where the total population is unchanged therefore making it easy to track and speak about percentages.

The authors defined an infectious individual as 'someone who has relatively high initial levels of pathogens on their hands at any given time', then is the individual infected too?

We understand how this point has caused confusion. We have included further clarity on the definitions and differences between 'infected' and 'infectious'. This change is only semantic and does not have an impact on any of our reported results.

In summary, 'infectious' (I1) individuals are the spreaders/contaminators of TUIs (they cannot be re-infected with their own germs / it's not of interest w.r.t assessing the risk

associated with TUI use). The 'infected' (I2) are individuals who start off as susceptible (S) and are exposed (E) to pathogens spread via TUI by I1. If the self-inoculation dose is sufficient, they become infected (I2), else they revert back to (S).

In order to assess the risk associated with TUI use, we are only interested in the ratio $R \sim (I2/I1)$ as this tells us the proportion of *newly* infected individuals to the original spreaders.

An exposed individual is someone who's contaminated hands after the interaction with a TUI, then is the individual infectious too?

By our definitions of infected and infectious, No. Moreover, re-deposition is not modelled in our framework. The contamination of hands/fingers after TUI use is calculated only to assess if a large enough dose is acquired for self-inoculation. The (I2) state behaves exactly as (S) in subsequent interactions (except that another self-inoculation won't be 'double-counted')

The authors should clearly state their definitions somewhere in the model description.

We thank the reviewer for highlighting this possible source of confusion. In our revised manuscript we try to clarify this complexity and how it is captured by our model both in the text and in our pseudo-code.

One of the model parameters is the disease prevalence, how is it actually used in the model? I suppose to initialize the population for the simulations.

That is correct. This parameter enables us to initiate different populations and model their evolution. We have re-worded the definition of this parameter to make it even more clear.

The prevalence of a disease is the proportion of a population infected at a given point in time. And the population under study is a population of infectious and susceptible individuals. But according to the authors' definition of infectious individuals (i.e., individuals with contaminated hands), infected individuals are not necessarily infectious. How do you deal with this?

The reviewer is correct in their understanding of our model. In our computer model we keep track of the state, location and interactions of all individuals and modify them at each simulation time step. This is how we can obtain deep insights about the network effects at play at multiple-TUI settings and how some individuals can statistically shelter susceptible individuals from contaminating their hands during touch interactions.

The model builds on a number of assumptions. Some of them sound questionable and would require further investigation.

We thank the reviewer for questioning some of our assumptions and provide responses to those below. It is however important to note that the main contribution of

our paper is the simulation framework which allows one to add/remove/edit these assumptions as required in a particular setting. In that sense our model is generalizable as it can adapt to different fomite interaction scenarios and for specific pathogen types.

Authors state that infected individuals 'remain infectious throughout the simulation' and that a simulation lasts one day. I find this assumption implausible.

This likely pertains to the confusion over 'infected' and 'infectious' individuals. Infected refers to a susceptible person who has self-inoculated. Infectious is the 'spreader' of pathogens (via TUIs) in this system. Therefore, it is the 'infectious' who continually spread pathogens (via coughing, sneezing etc.) throughout. The reviewer is somewhat correct in that this assumption is possibly over simplified. However, it is not totally unreasonable if we consider the peak shedding period, cough/sneeze rates and viral concentrations in droplets of (for example) an influenza sufferer. They are also unlikely to recover in a day.

The Discussion comments that the existence of such 'super' spreaders is either a limiting assumption of the model (or rather it highlights the requirement of this extreme case for fomite-mediated transmission to be plausible in the simulated scenarios presented)

Similarly, the self-inoculation timing of 20 min seems excessive as the effects of interactions with other surfaces are not negligible for such a long time window.

The reviewer is somewhat correct here as well in that this assumption is possibly over simplified. Individual touch behaviour may change significantly depending on the environment/scenario. If they are physically stood in a queue, they may be limited to what else they can touch and many of these touch events may not include the dominate index finger used (assumably) in TUI interaction. Moreover, many of these extra touch events will be face-touch events in the first instance.

What the 20 minutes assumption is saying (indirectly) is that the first 5 (on average) face touch events contribute the most to self-inoculation. After which the majority of pathogens are already transferred to the mucosal membranes have been removed through die-off or other surfaces being touched.

We have added further rationale (sec. Self-Inoculation) and discuss future plans for improvement in (Discussion).

Also, assuming susceptible, exposed, and newly infected individuals cannot deposit pathogens onto TUIs is a strong assumption, specific to the pathogens' nature and thus cannot be generalized.

The reviewer is somewhat correct here as well in that this assumption is possibly over simplified. However, the re-deposition of pathogens would effectively keep pathogens 'alive' in the simulation that would otherwise be removed in the current framework. For

the purposes of a risk assessment of TUI use, an underestimate of risk (fewer pathogens, less risk) is a preferred side to err on. Part of the reason for not modelling re-deposition is argued further in the revised text: see (Role of exposed (E) individuals) (Touchscreen Model). And we highlight it's shortcoming and future work plans in (Discussion).

Concerning the simulation parameters values, the authors are reporting 2 tables (Table 1 and Table A1), I would merge the 2.

We have merged the tables as advised.

Results analysis is weak and would merit a further discussion.

To strengthen the results analysis of our simulations we have included further discussion points both in the discussion section and also in the results section.

To conclude, I think that the authors have proposed a very general framework for modelling the transmission of infectious diseases through TUIs, presenting several unclear and uncertain points that would need further consideration. At this stage, no practical conclusions and/or recommendations can be drawn from it. Thus conclusions such as 'Results revealed that the required rate of cleaning/disinfecting of screens to effectively mitigate R can be inordinately high. This suggests that revised strategies or alternative methods should be considered.' sound not appropriate.

We thank the reviewer for their critical assessment of our manuscript and have tried to revise our conclusion section as to provide more clear and certain points about our contributions, results, and findings while also making some practical recommendations.

Namely, through our analysis and model simulations we have demonstrated several previously un-remarked results. These include the excessive amount of cleaning rates required, the importance of good timing, the network transport effect of pathogens from TUI to TUI and the possible shielding effect by individuals using the TUIs. These results provide a new dimension on the type of problems associated with disease transmission from touchscreen user interfaces to which we can only make educated suggestions and recommendations towards their mitigation. Using the framework developed we are hopeful that we will be able to test the effectiveness of mitigation strategies such as the ones proposed in the manuscript.

We hope that the modifications made and our responses to the reviewer's comments above are sufficient and acceptable for the publication of our paper to your journal.

Appendix B

Dear Dr Di Battista

On behalf of the Editors, we are pleased to inform you that your Manuscript RSOS-210625.R1 "Modelling disease transmission from touchscreen user interfaces" has been accepted for publication in Royal Society Open Science subject to minor revision in accordance with the Editors' comments below my signature.

We invite you to respond to the comments and revise your manuscript. Below the Editors' comments we provide additional requirements. Though further review of the manuscript is unlikely to be necessary after you prepare and resubmit the manuscript, final acceptance is dependent on these requirements being met. We provide guidance below to help you prepare your revision.

Please submit your revised manuscript and required files (see below) no later than 7 days from today's (ie 08-Jul-2021) date. Note: the ScholarOne system will 'lock' if submission of the revision is attempted 7 or more days after the deadline. If you do not think you will be able to meet this deadline please contact the editorial office immediately.

on behalf of Professor Enrico Bertuzzo (Associate Editor) and Mark Chaplain (Subject Editor)
openscience@royalsociety.org

Associate Editor Comments to Author (Professor Enrico Bertuzzo):

The authors revised the manuscript convincingly and it can be accepted (almost) as is.

The last 5 rows of the manuscript are out of the scope of the paper and I am asking the authors to delete them when preparing the final files. Moreover, please note that all references are gone in the version I am reading.

The last 5 lines of the conclusion have been removed. We agree that this may be a stretch beyond the scope of the paper.

The references did not appear in the reviewer version owing to the references being uploaded in a separate .bib file which did not compile with bibtex on the online system. We have included a final version that has all references embedded into the .tex main source file.

Pag 2, line 51 "On the other hand"

'One' has been changed to 'On'

===ZENODO DEPOSITION===

At this stage, we ask that you please archive your GitHub code within the Zenodo repository: <https://guides.github.com/activities/citable-code/>. By doing this, a formal, citable DOI will be associated with your data record, and an open license (CC-BY preferred) can be applied to your data. We would then ask that you please update your data availability statement to read as:

"Data and relevant code for this research work are stored in GitHub: [GitHub URL here] and have been archived within the Zenodo repository: <https://doi.org/zenodo.....> [ref number].

A github repo has been setup and Zenodo repository with doi number.

===PREPARING YOUR MANUSCRIPT===

- one version identifying all the changes that have been made (for instance, in coloured highlight, in bold text, or tracked changes);
- a 'clean' version of the new manuscript that incorporates the changes made, but does not highlight them. This version will be used for typesetting.

===PREPARING YOUR REVISION IN SCHOLARONE===

Journal Name: Royal Society Open Science

Journal Code: RSOS

Online ISSN: 2054-5703

Journal Admin Email: openscience@royalsociety.org

Journal Editor: Andrew Dunn
Journal Editor Email: openscience@royalsociety.org
MS Reference Number: RSOS-210625.R1
Article Status: SUBMITTED
MS Dryad ID: RSOS-210625.R1
MS Title: Modelling disease transmission from touchscreen user interfaces
MS Authors: Di Battista, Andrew; Christos, Nicolaidis; Georgiou, Orestis
Contact Author: Andrew Di Battista
Contact Author Email: andrew.di.battista@ultraleap.com
Contact Author Address 1:
Contact Author Address 2:
Contact Author Address 3:
Contact Author City: Bristol
Contact Author State:
Contact Author Country: United Kingdom of Great Britain and Northern Ireland
Contact Author ZIP/Postal Code: BS2 0EL
Keywords: fomite, touchscreen, disease transmission, stochastic model
Abstract: The extensive use of touchscreens for all manner of human-computer interactions has made them plausible instruments of touch-mediated disease transmission. To that end, we employ stochastic simulations to model human-fomite interaction with a distinct focus on touchscreen interfaces. The timings and frequency of interactions from within a closed population of infectious and susceptible individuals was modelled using a queuing network. A pseudo-reproductive number R was used to compare outcomes under various parameter conditions. We then apply the simulation to a specific real-world scenario; namely that of airport self check-in and baggage drop. A counter-intuitive result was that R decreased with increased touch rates required for touchscreen interaction. Additionally, as one of few parameters to be controlled, the rate of cleaning/disinfecting screens plays an essential role in mitigating R , though alternative technological strategies could prove more effective. The simulation model developed provides a foundation for future advances in more sophisticated fomite disease-transmission modelling.
EndDryadContent